# Impact and Advances in the Role of Bacterial Extracellular Vesicles in Neurodegenerative Disease and Its Therapeutics

**DOI:** 10.3390/biomedicines11072056

**Published:** 2023-07-21

**Authors:** Ashok Iyaswamy, Kejia Lu, Xin-Jie Guan, Yuxuan Kan, Chengfu Su, Jia Liu, Ravindran Jaganathan, Karthick Vasudevan, Jeyakumari Paul, Abhimanyu Thakur, Min Li

**Affiliations:** 1Mr. & Mrs. Ko Chi-Ming Centre for Parkinson’s Disease Research, School of Chinese Medicine, Hong Kong Baptist University, Hong Kong SAR, China; lukejia-8259@hkbu.edu.hk (K.L.); guanxinjie924@163.com (X.-J.G.); kanyx0827@163.com (Y.K.); 20481403@life.hkbu.edu.hk (C.S.); liujiatheone@hotmail.com (J.L.); 2Department of Biochemistry, Karpagam Academy of Higher Education, Coimbatore 641021, India; 3Preclinical Department, Faculty of Medicine, Royal College of Medicine Perak, Universiti Kuala Lumpur, Ipoh 30450, Malaysia; jravimicro@gmail.com; 4Department of Biotechnology, REVA University, Bangalore 560064, India; karthick.1087@gmail.com; 5Department of Physiology, Dr. ALM PG Institute of Basic Medical Sciences, University of Madras, Chennai 600005, India; jeyakumari.p.s@gmail.com; 6Pritzker School of Molecular Engineering, Ben May Department for Cancer Research, University of Chicago, Chicago, IL 60637, USA

**Keywords:** bacterial extracellular vesicles, therapeutics, neurodegenerative disease, Alzheimer’s disease, nanocarriers

## Abstract

Bacterial Extracellular Vesicles (BEVs) possess the capability of intracellular interactions with other cells, and, hence, can be utilized as an efficient cargo for worldwide delivery of therapeutic substances such as monoclonal antibodies, proteins, plasmids, siRNA, and small molecules for the treatment of neurodegenerative diseases (NDs). BEVs additionally possess a remarkable capacity for delivering these therapeutics across the blood–brain barrier to treat Alzheimer’s disease (AD). This review summarizes the role and advancement of BEVs for NDs, AD, and their treatment. Additionally, it investigates the critical BEV networks in the microbiome–gut–brain axis, their defensive and offensive roles in NDs, and their interaction with NDs. Furthermore, the part of BEVs in the neuroimmune system and their interference with ND, as well as the risk factors made by BEVs in the autophagy–lysosomal pathway and their potential outcomes on ND, are all discussed. To conclude, this review aims to gain a better understanding of the credentials of BEVs in NDs and possibly discover new therapeutic strategies.

## 1. Introduction

Neurodegenerative diseases (NDs) are a group of progressive disorders that affect the central nervous system and result in the gradual degeneration and mortality of neurons. Alzheimer’s disease (AD), Parkinson’s disease (PD), Huntington’s disease (HD), multiple sclerosis (MS), and amyotrophic lateral sclerosis (ALS) are among these conditions of NDs. AD International estimates that 50 million people worldwide are afflicted by dementia alone [1]. Consequently, NDs pose a significant threat to public health and a significant burden on society [1,2]. Despite significant efforts to develop treatments, no known definite cures for NDs are presently available [3].

Researchers have begun to investigate the function of bacterial extracellular vesicles (BEVs) in the pathogenesis of NDs in recent years [4,5]. BEVs are typically 50–200 nm in diameter, and they are composed of a lipid bilayer with a variety of proteins and nucleic acids (Figure 1). The cargo of BEVs can vary depending on the bacterial species and the growth conditions. They are like extracellular vesicles (EVs) released by eukaryotic cells, but they contain a different set of cargo molecules. Major EVs include exosomes and microvesicles, which have also been tremendously studied as theranostic agents and found to play crucial role in various physiological and pathological conditions [6,7,8]. BEVs are thought to play a role in bacterial communication, pathogenesis, and immunity. The biogenesis of BEVs is different from the biogenesis of exosomes and microvesicles [4,5]. BEVs are formed by the budding of vesicles from the outer membrane of bacteria. Exosomes are formed by the budding of vesicles from multivesicular bodies (MVBs) in eukaryotic cells. Microvesicles are formed by the direct outward budding of the plasma membrane of eukaryotic cells [9]. It has been found that BEVs serve an essential role in bacterial communication, adaptation, and virulence [10]. In addition, recent studies indicate that BEVs may also add to the development and progression of NDs, potentially presenting novel therapeutic targets [5,10].

The potential function of BEVs in NDs has been the subject of intensive research over the past few years [11]. According to studies, BEVs can play both offensive and defensive functions in the context of NDs, including regulation of immune responses, modulation of the microbiome–gut–brain axis, and induction of autophagy–lysosomal pathway dysfunction [5,11,12], as shown in Figure 1. In addition, recent advancements in BEV isolation, characterization, and engineering have paved the way for the development of novel therapeutic strategies [4,12].

This review’s objective is to provide an overview of the current literature on the role of BEVs in NDs and to investigate recent developments in the field of BEV therapeutics. We utilized different databases of biomedical literatures such as PubMed, Embase, Cochrane library, and Scopus, and we searched the following keywords: Neurodegeneration, Autophagy, Alzheimer’s disease, Nanocarriers, Extracellular vesicles, Exosomes, Therapeutics, NDs, ALP inducers, Parkinson’s disease, Brain delivery and BEVs. Specifically, this analysis seeks to focus on the offensive and defensive functions of BEVs in ND pathogenesis. It further elaborates the critical BEV networks in the microbiome–gut–brain axis and their role in ND. It discusses the roles of BEVs in the neuroimmune system and their interaction with ND. It intends to explore the risk factors of BEVs in the autophagy–lysosomal pathway and their possible effects on ND. It emphasizes the potential uses of BEVs in developing new drugs for ND. It suggests future prospects of the study for the role of BEVs in ND. This review seeks to contribute to a better understanding of the potential function of BEVs in NDs and to identify new therapeutic intervention strategies.

## 2. Offensive and Defensive Roles of Bacterial Extracellular Vesicles in Neurodegenerative Disease

Latest research has shown that BEVs [13] can cause neuroinflammation and affect neuronal function [14], which points to the potential neurotoxicity of these particles in the context of ND [14,15]. BEVs tend to yield the inflammatory cytokines and chemokines after activating microglia and astrocytes with virulence factors such lipopolysaccharides, peptidoglycans, and proteins [16,17,18]. AD, PD, and MS, among others, have all been associated with the activation of neuroinflammatory pathways [15,18].

BEVs generated from *Pseudomonas aeruginosa* were discovered in a recent study to induce inflammation and mortality of dopaminergic neurons in the substantia nigra [14,15,19]. Inflammation and mortality of dopaminergic neurons in the substantia nigra are two defining hallmarks of PD [14,18]. It was discovered that BEVs generated from Escherichia coli caused neuronal death and impaired memory in an AD mice model [18,19]. By inducing neuroinflammation and impairing neuronal function, BEVs may contribute to the pathogenesis of NDs, according to these findings [19,20]. 

In addition to their potential neurotoxicity [20], it has been demonstrated that BEVs serve a protective role in NDs [20]. Several studies have demonstrated, for instance, that BEVs can exert neuroprotective and immunomodulatory effects [20,21]. Specifically, it has been demonstrated that BEVs from commensal gut bacteria enhance cognitive function and reduce neuroinflammation in mouse models of NDs [21,22]. These results suggest that BEVs may also have therapeutic applications for the treatment of NDs [21,22]. The neurotoxicity of BEVs necessitates caution in their use as therapeutics, but their neuroprotective and immunomodulatory properties present opportunities for the development of novel treatments for NDs [13,21]. To thoroughly comprehend the mechanisms underlying the offensive and defensive roles of BEVs in NDs, additional research is required [13,20].

Recent research suggests that BEVs can also play a defensive function in NDs due to their neuroprotective and immunomodulatory properties [20,21]. Multiple studies have demonstrated, for instance, that BEVs derived from specific bacterial strains can improve neuronal survival and function in ND [22]. In the mouse model of AD, Haney et al. [23] found that BEVs from the probiotic *Lactobacillus rhamnosus* GG could reduce amyloid-beta (A) deposition and enhance cognitive function [24]. Few research studies have demonstrated that BEVs from *Bifidobacterium infantis* could reduce inflammation and oxidative stress in a mouse model of PD [25], resulting in enhanced motor function [21,22,25].

BEVs have been shown to possess immunomodulatory properties in the context of NDs, in addition to their neuroprotective effects [18,25]. By modulating the gut–brain axis [26], one study found that BEVs from *Akkermansia muciniphila* could reduce neuroinflammation and enhance cognitive function in a mouse model of PD [18,27]. Similarly, another study demonstrated that BEVs from Lactobacillus plantarum PS128 modulated microglial activity to enhance cognitive function and reduce neuroinflammation in an AD mouse model [25,28]. These studies suggest that BEVs may serve a dual role in NDs by possessing both offensive and defensive characteristics [18,28]. Some BEVs can induce neuroinflammation and impede neuronal function, whereas others can prevent diseases and modulate the immune system [21,24]. To thoroughly comprehend the mechanisms underlying the defensive properties of BEVs and to investigate their potential as therapeutic agents for NDs, additional research is required.

## 3. Critical Networks of Bacterial Extracellular Vesicles in the Microbiome–Gut–Brain Axis

The microbiome–gut–brain axis (MGBA) is a complex network of bidirectional communication between the gastrointestinal tract, the central nervous system (CNS), and the gut microbiota [22,29]. Recent evidence suggests that this axis regulates a variety of physiological and pathological processes, such as neuroinflammation and neurodegeneration [29,30]. The gut microbiota has a vast array of microorganisms inhabiting the human gastrointestinal tract, and it has been shown to influence brain function and behavior via multiple mechanisms [30]. These include the production of neurotransmitters and short-chain fatty acids, modulation of the immune system, and regulation of the hypothalamic–pituitary–adrenal axis [31,32]. Multiple NDs such as PD [31], AD [32,33], and MS have been linked to abnormalities in the gastrointestinal microbiota in their pathogenesis [33,34,35]. Additionally, it has been demonstrated that BEVs produced by intestinal microbiota can cross the blood–brain barrier (BBB) and directly affect the CNS function [25,34]. BEVs from the gut commensal *Bacteroides fragilis* have been shown to facilitate the differentiation and maturation of oligodendrocytes, which are essential to produce myelin in the CNS [5,36]. In a mouse model of AD, BEVs from *Akkermansia muciniphila* have been shown to protect against neuroinflammation and cognitive decline [26,27,36]. 

The above discussed results point out that the MGBA plays an important role in the pathogenesis of NDs, and that BEVs produced by intestinal microbiota may represent a novel drug delivery system for such conditions [37]. It has also been demonstrated that the gut microbiome can influence brain function and behaviors via multiple mechanisms, including the production of neurotransmitters, regulation of the immune system, and modulation of the gut–brain axis signaling pathways [38]. BEVs, which are produced by numerous bacteria in the microbiome of the gut, have been identified as potential mediators of this communication between the gut and the brain [5,38]. The effects of BEVs on the microbiome–gut–brain axis and their potential function in NDs have been studied and discussed in animal models [21]. In cell cultures and mouse models, BEVs from the gastrointestinal microbiome of PD patients were able to induce alpha–synuclein aggregation, which is a hallmark of PD pathology [18,25]. In a mouse model of AD, it was observed that BEVs from a specific gut bacterium, *Akkermansia muciniphila*, reduced neuroinflammation and enhance cognitive function [39]. 

Another study identified a group of BEVs produced by gut bacteria that could cross the BBB as a result, penetrating the brain, modulating the immune system [40], and potentially playing a significant role in NDs [20,30]. These studies indicate that BEVs can play a significant role in the communication between the gastrointestinal microbiome and the brain, and that their dysregulation may contribute to the development and progression of NDs [39,40]. The mechanisms underlying the effects of BEVs on the microbiome–gut–brain axis and their potential as therapeutic targets for NDs require additional study [30,39]. Among the NDs, a research study has shown that in AD brain, microglial activation contributes to amyloid-beta deposition and neuronal damage [41]. In addition, in PD few research studies have shown that in the brain T cells infiltrate the substantia nigra and promote neuroinflammation [42,43]. Moreover, in multiple sclerosis, few studies have shown that dysbiosis and gut-derived molecules contribute to neuroinflammation and disease progression [44,45,46].

## 4. Role of Bacterial Extracellular Vesicles in Neuroimmune System and Their Crosstalk

The neuroimmune system, which is made up of interactions between the neurological system and the immune system, is critical in NDs [47,48]. This neuroimmune system helps keep the homeostasis in balance. If this balance is distraught, it can lead to chronic inflammation, damage to neurons, and eventually NDs [41,42]. In terms of NDs, the neuroimmune system is made up of immune cells like microglia and astrocytes that become active when there is neuroinflammation [43]. When these cells become active, they release cytokines and chemokines that cause more inflammation and damage to neurons [43,44]. Peripheral immune cells, such as T cells and monocytes, can also promote neuroinflammation by crossing the BBB and entering the central nervous system [44,45].

Recent studies have shown how important the microbiome–gut–brain axis is in NDs and how it affects the neuroimmune system [49]. Dysbiosis, which is an imbalance in the gut microbiome, has been linked to the development of NDs [49,50]. This could be because small molecules from the gut, like lipopolysaccharides, affect the immune system [16,41]. The latest studies have looked at how BEVs and the neuroimmune system interact, which shows how BEVs might be able to change the immune response in NDs [16,20,40]. For example, BEVs made from the gut bacteria *Bacteroides fragilis* suppressed the immune response in a mouse model of multiple sclerosis [51]. It was found that the BEVs helped regulatory T cells grow; regulatory T cells are very important for calming down immune responses and preventing autoimmunity [52]. Table 1 lists the major bacterial source of EVs involved in the neuroimmune system’s interactions in the pathogenesis and therapeutics of NDs.

BEVs from the gut bacteria *Akkermansia muciniphila* were demonstrated to diminish neuroinflammation in a mouse model of PD [26,27]. It was found that the BEVs decreased the number of pro-inflammatory cytokines in the brain and increased the number of anti-inflammatory cytokines [16,18]. This suggests that BEVs have a neuroprotective effect. A study looked at how BEVs from the gut bacteria Bifidobacterium bifidum might affect the immune system in an AD animal model [24,46]. The researchers found that giving BEVs to the mice led to less inflammation in the brain and better brain function [46]. These studies show that BEVs may be able to change the immune response in NDs, which means they may be a good way to treat these diseases [20,49]. But more research is needed to fully understand the mechanisms behind these effects and to figure out the best ways to use BEVs as medicines.

## 5. Risk Factors of Bacterial Extracellular Vesicles in Autophagy–Lysosomal Pathway

The autophagy–lysosomal pathway (ALP) is a cellular process that gets rid of damaged organelles, misfolded proteins, and invading pathogens by breaking them down and recycling them [55,56]. Dysregulation of the ALP has been linked to ND. Studies of bacterial infections have shown that various types of EVs are released, including exosomes and microvesicles. The composition of the EV cargo can vary depending on the infection and cell type, and this can ultimately impact the host immune response and bacterial growth [57]. Figure 2 depicts the autophagy-related pathways that employs LC3 conjugation to membrane endocytic and phagocytic vesicles and their effect on EVs’ release. 

The ALP has three main types: macroautophagy, microautophagy, and chaperone-mediated autophagy, as shown in Figure 3. Autophagy or macroautophagy has been studied the most and it involves the formation of autophagosomes that engulf cytoplasmic parts and fuse with lysosomes to break them down [58,59].

ALP is very important in removing toxic proteins, like alpha-synuclein in PD, Aβ, and phospho tau in AD, from building up in the brain and trying to clear this toxic proteins when activated. Also, abnormal regulation of the ALP has been linked to the activation of inflammatory pathways and oxidative stress, both of which contribute to the development of NDs [45,60]. BEVs may be able to promote the ALP in different NDs, such as PD and AD [16,18,20]. Recent studies have shown that by controlling or promoting the ALP using BEVs, suggesting that BEVs could be a promising therapeutic target for treating NDs focusing on macroautophagy, chaperone-mediated autophagy, and microautophagy [46,53,61].

The ALP is very important for keeping cells in balance, and its malfunction has been linked to the development of several NDs. More research is needed to fully understand how the ALP works in NDs and to investigate the possibility that EVs, BEVs, or other external agents that could be used as a therapeutics to control the ALP are required for the present situation in the treatment of NDs [9,62,63]. As per the previous studies, EVs as brain delivery nanocarriers or other phytochemicals has been reported that may influence the autophagy–lysosomal pathway, in turn helping or keeping cells in balance by getting rid of damaged organelles and protein clusters [9,64,65]. Autophagy is a strictly regulated process that involves the creation of autophagosomes, which are double-membrane vesicles [56,66]. These vesicles take in the toxic cytoplasmic materials and send them to lysosomes by fusing them to form autolysosome to clear or break down the engulfed proteins or organelles [67,68]. There are many hydrolytic enzymes in lysosomes that can break down the contents of autophagosomes into nutrients that can be used to make energy and change the shape of cells [69,70].

Studies have shown that BEVs can mess up the autophagy–lysosomal pathway, which makes it harder for cells to get rid of waste and causes toxic aggregates to build up [70,71]. For example, a recent study showed that BEVs made from *Porphyromonas gingivalis*, a pathogenic oral bacterium linked to AD, could stop autophagy by stopping lysosomes from becoming acidic and stopping autophagosomes from breaking down [71,72,73]. BEVs made from *Bacteroides fragilis*, a common gut bacterium that can change the immune system, could stop autophagy in dendritic cells by stopping the fusion of autophagosomes and lysosomes [30,46,73]. On the other hand, some studies have shown that BEVs may be playing a protective role in the autophagy–lysosomal pathway by assisting to make new lysosomes and speeding up autophagic flux [46,69,71]. For example, a recent study showed that BEVs made from Lactobacillus acidophilus, a probiotic bacterium with anti-inflammatory properties, could improve autophagy flux by increasing lysosomal biogenesis and promoting lysosomal acidification [74,75]. It was also found that BEVs made from *Akkermansia muciniphila* linked to better metabolic health could speed up the removal of misfolded proteins in a mouse model of PD by activating the autophagy–lysosomal pathway [27,75,76]. In ND, the connection between BEVs and the autophagy–lysosomal pathway is complicated and needs to be investigated more [27,75,77]. The possibility that BEVs could interfere with or improve this important way for cells to get rid of waste could have big effects on how ND start and how they can be treated.

## 6. Therapeutic Applications of Bacterial Extracellular Vesicles

In east Asian countries, traditional Chinese medicine (TCM) makes use of herbal concoctions to treat NDs such as AD and PD for many years. However, BEVs can become an effective alternative for such treatment [63,72,75]. Depending on their source and the surrounding environment, exogenous vesicles have the potential to both trigger immune responses and induce anti-tumor responses; as a result, they may offer useful tools for the development of innovative cancer treatments [72,78].

Since BEVs can cross the blood–brain barrier, several recent studies have investigated how BEVs could be used to treat NDs [74]. They can change immune responses and protect neurons from damage [62,79]. Evidently, BEVs have the potential to be used as a drug delivery system to treat NDs [74]. BEVs can carry a wide range of cargo, such as proteins, nucleic acids, and small molecules. This makes them an ideal vehicle to deliver active agents to specific cells or parts of the brain [80]. In fact, several studies have shown that therapeutic cargo can be delivered by BEVs in nonclinical models of NDs [80,81,82]. For example, one research study used BEVs made from mesenchymal stem cells to deliver miR-133b to neurons in a mouse model of AD [80,81] and this study illustrated the tremendous output in cognitive function and lowered the levels of amyloid beta [80,81,82].

It has been repeatedly found that BEVs can not only change the immune system but also protect nerve cells, which makes them a very useful drug delivery system for treating NDs [82]. Reducing neuroinflammation, a characteristic feature of many NDs, has been demonstrated as an effect of BEVs on the brain’s immune system [75,81]. Another very interesting and important factor is that BEVs protect neurons from damage by lowering oxidative stress and helping neurons stay alive [75,81].

It is important to remember that while using BEVs to treat NDs has much potential, there are also risks and challenges. For example, BEVs may interfere with the way lysosomes work and slow down the waste removal system of the cell, which can certainly cause toxic proteins and other cellular debris to build up and cause different kinds of issues [83,84]. Another challenge is the lack of information. While the research has shown some promising results, there is a major possibility that the lack of substantial information about the working of the pharmacokinetics, its safety, and effectiveness of BEVs in humans, especially when it comes to long-term use and possible immune responses, can produce some side effects [63,77,84].

Even with these certain and uncertain challenges, the possible uses of BEVs in NDs are an interesting and intriguing area of research that needs more study and development [63,75]. With continued progress in our understanding of how NDs work and how BEVs affect these processes, BEVs may end up being a useful addition to our arsenal of neuroprotective and disease-modifying therapies [69]. BEVs are starting to look like a good way to treat diseases that damage nerve cells [75]. Recent preclinical studies on mouse models of NDs have shown that BEVs can change the immune response, lower neuroinflammation, and improve cognitive function [75,82]. Also, BEVs have been shown to protect neurons in the lab by preventing the buildup of misfolded proteins, encouraging autophagy, and improving lysosomal function [66,69,85].

On a practical note, several preclinical studies have investigated the possibility that BEVs could be used to deliver drugs to treat NDs [63,74,81]. BEVs used as drug delivery agents in model organisms have depicted some promising results. For example, BEVs were allowed to carry therapeutic agents like siRNA and drugs to specific brain cells [62,74,81]. The results showed that BEVs made from mesenchymal stem cells can deliver active siRNA that targets the tau protein in the brain and reduces tau phosphorylation and clumping in a mouse model of AD [62,81]. The use of BEVs to treat NDs in a clinical setting is still in its early stages [84]. Clinical trials and several other studies are being conducted and reports are mostly positive.

## 7. Future Outlook of Bacterial Extracellular Vesicles

In recent years, research on the role of BEVs in NDs has garnered considerable interest. Although there has been significant progress in our understanding of how BEVs might be used therapeutically, much remains to be learned, particularly regarding their interaction with the autophagy–lysosomal pathway. Additionally, further research is needed to explore the potential risks of using BEVs, such as the chance of eliciting an adverse immune response or unintended biological effects. The optimization of BEV delivery to the brain is another area of essential future research. Currently, there is little knowledge of the various factors impacting BEV uptake and distribution, both of which can have a significant influence on therapeutic efficacy. Nanotechnology and other delivery technologies may be useful in increasing the target specificity and efficacy of BEV delivery. In addition, standard protocols for isolating and documenting BEVs need to be developed, establishing a uniform approach for the description and comparison of findings from different studies. The current understanding of how BEVs affect NDs and their inner workings is also insufficient. Although various studies have shown that BEVs can modify the immune system, promote the survival of neurons, and eliminate misfolded proteins, the specific molecular mechanisms responsible for these effects remain largely elusive. This underscores the need for extensive safety testing and validation of BEVs before their clinical implementation.

## 8. Conclusions

BEVs possess the potential to make a significant contribution to ND and their therapeutics. Recent studies have indicated that these vehicles have immuno-modulatory and neuroprotective properties, which may prove beneficial in the treatment of NDs. Furthermore, it has been observed that BEVs have the capacity to traverse the BBB and target parts of the brain, making them an attractive drug delivery system for NDs. However, there remain impediments and limitations to consider when studying and opting for BEVs in NDs. For example, the lack of standardization in BEV isolation and characterization processes can lead to disparate data and impede the comparison of studies. In addition, further research is needed to deepen our understanding of the mechanisms through which BEVs exert their effects. To maximize their therapeutic potential for NDs, future investigations should focus on mitigating these challenges and limitations. To ascertain their safety and effectiveness in humans, further examination into the most suitable dosage, frequency, and duration of BEV treatment is necessary. Moreover, further research is also mandated to further our knowledge of the underlying mechanisms of action of BEVs and to discern the BEVs with the most potential for treating NDs. Ultimately, BEVs provide a promising therapeutic avenue for NDs yet a valid comprehension of their magnitude and the surmounting of these restrictions of use necessitate additional research. Taken as a whole, BEVs are an intriguing research topic that deserves further exploration.

## Figures and Tables

**Figure 1 biomedicines-11-02056-f001:**
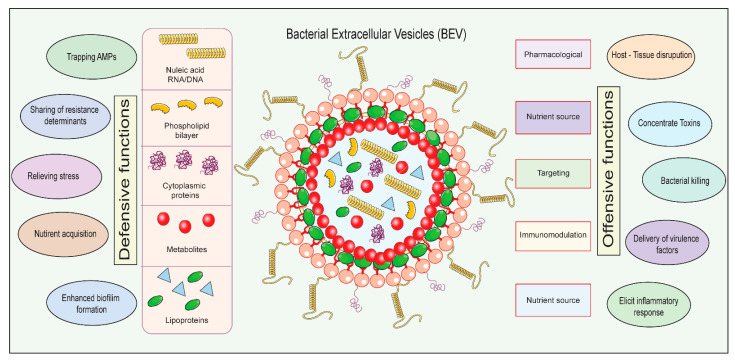
A schematic showing the structure of bacterial extracellular vesicle (BEV) and their components with different functions, along with offensive and defensive roles of BEV in neurodegenerative disease and its prospective therapeutic applications.

**Figure 2 biomedicines-11-02056-f002:**
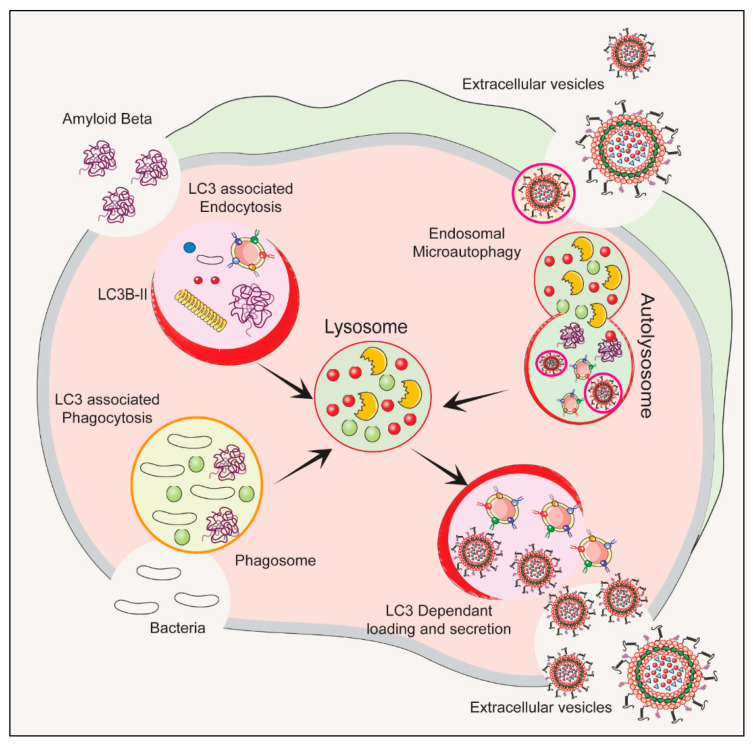
The process of initiation, nucleation and maturation of autophagy including the autophagosome formation and autophagy flux for the formation of autolysosome associated with LC3 Regulation of autophagy is dependent on the LC3 associated endocytosis, LC3 associated phagocytosis, Endosomal microautophagy and LC3 dependent loading and secretion of extracellular vesicles (EVs). The release of EVs and autophagy are two complementary mechanisms that cells use to eliminate amyloids and protein aggregates. EVs such as exosomes bud from late endosomes, which are themselves derived from multivesicular bodies (MVBs), which can either be released extracellularly or degraded in lysosomes. Autophagy is a cellular process in which cytosolic cargoes are sequestered into autophagosomes, which then fuse with lysosomes for degradation.

**Figure 3 biomedicines-11-02056-f003:**
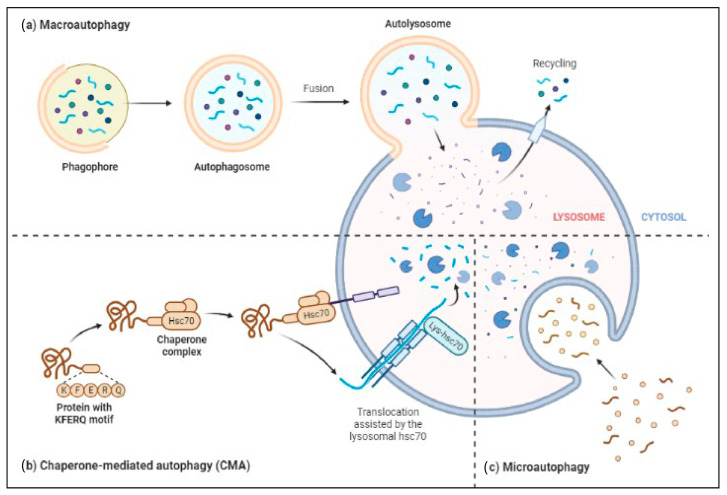
Three major types of autophagy–lysosomal pathways in NDs and other brain diseases are macroautophagy, chaperone-mediated autophagy, and microautophagy. Macroautophagy process degrades or eradicates the damaged cell organelles, unused protein, and toxic proteins by generating autophagosomes and fuse with lysosome. Chaperone-mediated autophagy degrades the unused proteins, and the misfolded proteins and intracellular toxic proteins are proteolytically degraded directly in lysosomes by translocating via the lumen of lysosomes. Microautophagy is a nonselective degradative process by accumulating the cytoplasmic contents directly to the lysosome by forming the endosomes.

**Table 1 biomedicines-11-02056-t001:** Studies that describe how BEVs and the neuroimmune system interact in the pathogenesis and therapeutics of NDs.

BEV Source	Disease Type	Key Findings	Ref.
*Bifidobacterium bifidum*	AD	BEVs reduced neuroinflammation and improved cognitive function	[24,25,46]
*Akkermansia muciniphila*	PD	BEVs reduced neuroinflammation by decreasing pro-inflammatory cytokine levels while raising anti-inflammatory cytokine levels	[26,27,53]
*Bacteroides fragilis*	MS	BEVs promoted the expansion of regulatory T cells, dampening immune responses and preventing autoimmunity	[24,52,54]

## Data Availability

The data presented in this study are available on request from the corresponding author.

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
