# Peer review of "Impact and Advances in the Role of Bacterial Extracellular Vesicles in Neurodegenerative Disease and Its Therapeutics"

_biomedicines, 2023, doi:10.3390/biomedicines11072056_

Round 1

Reviewer 1 Report

At first, the present manuscript was submitted for Special Issue "Amyloid beta, Tau, and alpha-Synuclein aggregates in the Pathogenesis, Prognosis, and Therapeutics for Neurodegenerative Diseases". However, the review was not about aggregates. Was it OK that the present submission was for this special issue? 

On the other hand, there were some minor comments.

In the present manuscript, the authors used one abbreviation "NDs" for neurodegenerative diseases and neurodegenerative disorders. Was it needed to use both neurodegenerative diseases and neurodegenerative disorders? Did the authors use these two words separately? If no, they they would be better to unify with either.

The section numbers were all 1. Renumber them.

In the section, offensive and....; "(Kim et al., 202; Kumar et al., 2020)" was not properly cited. They should be corrected and added in References with proper numbers of references.

In addition, the authors wrote "Wang et al. [21] demonstrated that...."; however, [21] was not by Wang et al. and Wang et al. was absent in the reference section. It should be corrected and the proper reference should be cited.

In the present review article, it was not clear why the authors reviewed the effects of bacterial extracellular vesicles in autophagy lysosomal pathway. The pathway has been surely linked to neurodegenerative diseases, but various factors were linked to the diseases. Why did the authors focus on autophagy lysosomal pathway in the present review article? It should be clearly explained.

The abbreviations used in the present manuscript were sometimes explained as full-names. When the authors used at first the abbreviations should be written with full-names, and thereafter abbreviations alone should be used.

The quality of English was fine.

Author Response

Point-by-point response letter to reviewer’s comments

We thank the reviewers for providing the valuable comments for revision. We have addressed each comment carefully point by point and made revisions accordingly.

Reviewer-1

At first, the present manuscript was submitted for Special Issue "Amyloid beta, Tau, and alpha-Synuclein aggregates in the Pathogenesis, Prognosis, and Therapeutics for Neurodegenerative Diseases". However, the review was not about aggregates. Was it OK that the present submission was for this special issue? 

Response: Thank you for your valuable comments. We highly appreciate your time and comments to improve our manuscript. We agree with the reviewer comments on the special issue topic, but this manuscript well relates to the scope of the special issue elucidating the recent advancement in the therapeutics for neurodegenerative diseases.

On the other hand, there were some minor comments.

In the present manuscript, the authors used one abbreviation "NDs" for neurodegenerative diseases and neurodegenerative disorders. Was it needed to use both neurodegenerative diseases and neurodegenerative disorders? Did the authors use these two words separately? If no, they they would be better to unify with either.

Response: We appreciate the reviewer’s comment. As suggested, we have corrected in the whole manuscript for the abbreviation.

The section numbers were all 1. Renumber them.

Response: We appreciate the reviewer’s comment. As suggested, we have corrected and renumbered the subsections in the manuscript.

In the section, offensive and....; "(Kim et al., 202; Kumar et al., 2020)" was not properly cited. They should be corrected and added in References with proper numbers of references.

Response: We appreciate the reviewer’s comment. As per your suggestion, we have corrected the reference and removed the name format as per the reference style.

In addition, the authors wrote "Wang et al. [21] demonstrated that...."; however, [21] was not by Wang et al. and Wang et al. was absent in the reference section. It should be corrected and the proper reference should be cited.

Response: We appreciate the reviewer’s comment. As suggested, we have corrected reference and removed the name as per the reference numbers.

In the present review article, it was not clear why the authors reviewed the effects of bacterial extracellular vesicles in autophagy lysosomal pathway. The pathway has been surely linked to neurodegenerative diseases, but various factors were linked to the diseases. Why did the authors focus on autophagy lysosomal pathway in the present review article? It should be clearly explained.

Response: We appreciate the reviewer’s comment. As suggested, we have included the relevant texts in the manuscript. Even though ALP and NDs are corelated in several aspects, the present scenario on development of therapeutics for NDs targeting ALP is elusive, so we try to find new and novel approaches to target ALP for the treatment of AD, PD and other NDs. To demonstrate good understanding on BEVs for brain delivery in NDs, we prepared this review article. This review seeks to contribute to a better understanding of the potential function of BEVs in NDs and to identify new therapeutic intervention strategies.

The abbreviations used in the present manuscript were sometimes explained as full-names. When the authors used at first the abbreviations should be written with full-names, and thereafter abbreviations alone should be used.

Response: We appreciate the reviewer’s comment. As per your suggestion, All the abbreviations have been resolved and expanded at their first appearance in the text.

Reviewer 2 Report

I received a very interesting review article for review.
I read it with pleasure. Highly refined figures are its additional advantage.
The choice of topic is very up-to-date, and the researchers' approach is interesting.

I lacked methodology - even a few sentences about the criteria according to which literature was selected. What databases the authors browsed, and what keywords they used, which was a possible criterion for excluding the selected articles?
I noticed the wrong numbering of sections, number 1 appears everywhere.

I recommend only minor changes. 

Author Response

Point-by-point response letter to reviewer’s comments

We thank the reviewers for providing the valuable comments for revision. We have addressed each comment carefully point by point and made revisions accordingly.

Reviewer-2

I received a very interesting review article for review.
I read it with pleasure. Highly refined figures are its additional advantage.
The choice of topic is very up-to-date, and the researchers' approach is interesting.

Response: Thank you for your valuable comments. We highly appreciate your time and comments to improve our manuscript.

I lacked methodology - even a few sentences about the criteria according to which literature was selected. What databases the authors browsed, and what keywords they used, which was a possible criterion for excluding the selected articles?

Response: We appreciate the reviewer’s comment. As suggested, the used keywords have been included in the manuscript as follows: We utilized different databases of biomedical literatures such as PubMed, Embase, Cochrane library, and Scopus; with keywords: Neurodegeneration, Autophagy, Alzheimer’s disease, Nanocarriers, Extracellular vesicles, Exosomes, Therapeutics, NDs, ALP inducers, Parkinson’s disease, Brain delivery and BEVs.

I noticed the wrong numbering of sections; number 1 appears everywhere.
I recommend only minor changes. 

Response: We appreciate the reviewer’s comment. As suggested, we have corrected and renumbered the subsections in the manuscript.

Round 2

Reviewer 1 Report

The authors sincerely responded the reviewers' comments and improved the manuscript. The revised manuscript might be acceptable for publication if the editors would judge to match the scope of the special issue.

There were some spelling errors.